# Impact of Acute Myeloid Leukemia Cells on the Metabolic Function of Bone Marrow Mesenchymal Stem Cells

**DOI:** 10.3390/ijms26178301

**Published:** 2025-08-27

**Authors:** Helal Ahmed, Pradeep Kumar Patnana, Yahya S. Al-Matary, Maren Fiori, Jan Vorwerk, Marah H. Ahmad, Eva Dazert, Lorenz Oelschläger, Axel Künstner, Bertram Opalka, Nikolas von Bubnoff, Cyrus Khandanpour

**Affiliations:** 1Medical Department A, University Hospital Münster, 48149 Münster, Germany; 2Department of Hematology and Oncology, University Cancer Center Schleswig-Holstein (UCCSH), University Hospital Schleswig-Holstein, 23538 Lübeck, Germany; 3Faculty of Medicine and Health Sciences, Amran University, Amran 00967, Yemen; 4Clinic for Oral and Maxillofacial Surgery, University Medical Center Göttingen, 37075 Göttingen, Germany; yahya.al-matary@med.uni-goettingen.de; 5Department of Hematology and Stem Cell Transplantation, West German Cancer Center Essen-Münster (WTZ), University Hospital Essen, 45147 Essen, Germany; 6Division of Immunobiology, Cincinnati Children’s Hospital, Cincinnati, OH 45229-3039, USA; 7Medical Systems Biology Group, Lübeck Institute of Experimental Dermatology, University of Lübeck, 23538 Lübeck, Germany

**Keywords:** acute myeloid leukemia, bone marrow mesenchymal stromal cell, metabolism

## Abstract

Acute myeloid leukemia (AML) proliferation is significantly influenced by the interactions between leukemia blasts and the bone marrow (BM) microenvironment. Specifically, bone marrow mesenchymal stem cells (BMSCs) derived from AML patients (AML-MSCs) are known to support leukemia growth and facilitate disease progression. Studies have demonstrated that the transfer of mitochondria from MSCs to AML blasts not only aids in disease progression but also contributes to chemotherapy resistance. Furthermore, BM stromal cells can trigger a metabolic shift in malignant cells from mitochondrial respiration to glycolysis, which enhances both growth and chemo-resistance. This study focuses on identifying transcriptional and metabolic alterations in AML-MSCs to uncover potential targeted therapies for AML. We employed RNA sequencing and microarray analysis on MSCs cocultured with leukemic cells (*MLL-AF9*) and on MSCs isolated from both non-leukemic and *MLL-AF9* leukemic mice. The Gene Set Enrichment Analysis (GSEA) indicated a significant downregulation of gene sets associated with oxidative phosphorylation and glycolysis in AML-MSCs. Furthermore, coculture of MSCs from wild-type mice (WT-MSCs) and a healthy donor individual (HD-MSCs) with AML cells demonstrated reduced oxidative phosphorylation and glycolysis. These metabolic changes were consistent in AML-MSCs derived from both leukemic mice and patients. Our results indicate that AML cells diminish the metabolic capacity of MSCs, specifically targeting oxidative phosphorylation and glycolysis. These findings suggest potential metabolic vulnerabilities that could be exploited to develop more effective therapeutic strategies for AML.

## 1. Introduction

Acute myeloid leukemia (AML) is an aggressive hematological malignancy with high relapse rates and poor overall prognosis [1]. Bone marrow mesenchymal stem cells (MSCs), particularly those derived from AML patients (AML-MSCs), have been shown to significantly support leukemic cell growth and facilitate disease progression through direct cell–cell interactions and paracrine signaling, as demonstrated by Ahmed et al. and others [2,3,4,5,6,7]. AML cells strongly depend on mitochondrial oxidative phosphorylation (OXPHOS) to sustain their elevated metabolic demands. Emerging evidence indicates that AML cells acquire functional mitochondria from MSCs, enhancing their survival, self-renewal, and resistance to chemotherapy [8,9,10,11,12,13,14].

In addition to mitochondrial transfer, AML cells have been shown to reprogram stromal cells to secrete acetate, which serves as a substrate for the tricarboxylic acid (TCA) cycle and lipid biosynthesis in leukemic cells, further supporting their metabolic needs and proliferation [15]. Within the leukemic microenvironment, malignant cells frequently undergo a metabolic shift from oxidative phosphorylation (OXPHOS) to glycolysis, a phenomenon commonly referred to as the Warburg effect. This metabolic reprogramming is characterized by increased glucose uptake and lactate production, even in the presence of sufficient oxygen, leading to a reliance on glycolysis for energy generation rather than mitochondrial respiration. The Warburg effect provides several advantages to leukemic cells, including rapid ATP production, the generation of biosynthetic precursors necessary for cell growth and division, and the ability to maintain redox balance through increased NADPH production. Furthermore, by acidifying the tumor microenvironment through lactate secretion, leukemic cells can modulate immune cell function and evade immune surveillance. This metabolic adaptation also plays a crucial role in therapeutic resistance, as glycolytic cells tend to be less susceptible to mitochondrial-targeting drugs and oxidative stress induced by chemotherapy.

Despite substantial advances in understanding the metabolic crosstalk between AML cells and the stromal microenvironment, the metabolic profile of AML-associated stromal cells remains poorly characterized. To address this gap, we investigated the transcriptional and metabolic alterations in AML-MSCs to identify potential therapeutic targets to improve AML treatment outcomes.

Our findings revealed significant downregulation of hallmark gene sets related to OXPHOS and glycolysis in AML-MSCs, as determined by Gene Set Enrichment Analysis (GSEA). Furthermore, functional studies using Seahorse mito-stress assays demonstrated reduced oxidative phosphorylation and glycolytic activity in AML-MSCs. These data indicate a profound metabolic impairment in MSCs under the influence of leukemic cells, particularly within key metabolic pathways. Targeting these suppressed metabolic pathways in the stromal compartment may represent a novel therapeutic strategy for improving AML treatment efficacy and overcoming chemoresistance.

## 2. Results

### 2.1. Molecular Changes in MSCs Cocultured with AML Cells and in MSCs from Leukemic Versus Non-Leukemic Mice

Our previous work established that direct interactions between AML cells and MSCs derived from AML patients (AML-MSCs) contribute to enhanced leukemic cell proliferation, disease progression, and survival [7]. We hypothesized that AML cells impose metabolic constraints on MSCs, leading to functional alterations. To further explore the transcriptional modifications induced by leukemic cells, we isolated MSCs from healthy control (WT-MSCs) and leukemic (AML-MSCs) mice (Appendix A). These AML-MSCs were derived from leukemic mice expressing MLL–AF9 oncoproteins associated with the t(9;11) (p22;q23) translocation, a model extensively used for studying AML pathogenesis [16,17,18,19]. Control mice were sub-lethally irradiated and transplanted with healthy, non-transduced bone marrow (BM) cells, while AML was induced in recipient mice through secondary transplantation of leukemic BM cells. Following leukemia induction, mesenchymal stromal cells (MSCs) were isolated from both non-leukemic (WT-MSCs) and leukemic (MLL–AF9) mice [2]. Gene expression profiling using RNA microarrays revealed significant downregulation of gene sets associated with oxidative phosphorylation (OXPHOS) and glycolysis in AML-derived MSCs, as demonstrated by Gene Set Enrichment Analysis (GSEA) (Figure 1A,B).

To further elucidate AML-induced transcriptional alterations in vitro, WT-MSCs isolated from wild-type mice (Appendix A) were characterized by positive expression of MSC-specific markers (Sca-1, CD51, CD44, and CD29) and absence of hematopoietic markers (CD45 and CD31), as previously described [2]. These MSCs demonstrated differentiation capacity into adipocytes, osteocytes, and chondrocytes [2].

We subsequently cultured WT-MSCs either alone or in long-term coculture with MLL-AF9 cells and compared these to MSCs cultured with normal lineage-negative cells (Appendix A). Following sorting, RNA-seq analysis was performed, and GSEA confirmed a significant downregulation of gene sets related to OXPHOS and glycolysis in MSCs cocultured with AML cells (Figure 1C,D). A heatmap illustrating the scaled expression values of the top 20 genes from the OXPHOS and glycolysis pathways is shown in Figure 1E,F.

### 2.2. AML Cells Impair Metabolic Function of Murine MSCs

To validate the GSEA findings, we cocultured WT-MSCs with murine AML cells (C1498 cell line and MLL-AF9 cells) or normal Lin- cells for five days. Sorted MSCs (CD45– Sca-1 + CD51+) were analyzed for oxidative phosphorylation (OXPHOS) by measuring the oxygen consumption rate (OCR) and extracellular acidification rate (ECAR) using the Seahorse mito-stress assay. Additionally, glucose consumption and lactate secretion were measured from culture supernatants after three and five days, normalized to cell growth. The results demonstrated reduced OXPHOS in MSCs cocultured with both C1498 (Figure 2A,B) and MLL-AF9 cells (Appendix A), as well as decreased ECAR (Figure 2C,D and Appendix A), glucose uptake, and lactate production (Figure 2E and Appendix A). A concurrent reduction in the mitochondrial DNA copy number in MSCs cocultured with C1498 and MLL-AF9 cells (Figure 2F and Appendix A) further confirmed mitochondrial depletion. Similarly, MSCs cocultured with a murine AML cell line C1498 and human AML cell line (Kasumi and THP) showed decreased OXPHOS levels compared to those found in cocultures with Lin- cells (Appendix A). We further assessed reactive oxygen species (ROS) levels, mitochondrial mass, and mitochondrial membrane potential using flow cytometry. All three parameters were significantly reduced in MSCs cocultured with C1498 cells (Figure 2G–I and Appendix A). In addition, MSCs cocultured with C1498 cells display reduced proliferative capacity (Appendix A). Collectively, these findings are consistent with GSEA data, indicating that AML cells reduce OXPHOS and glycolytic activity in MSCs. These findings potentially align with reports that mitochondria transfer from MSCs to AML cells takes place through direct contact [13,14].

### 2.3. Impact of Acute Myeloid Leukemia Cells on the Metabolic Profile of Human Mesenchymal Stem Cells

To investigate whether the metabolic suppression observed in murine models extends to human systems, we conducted a series of experiments utilizing the human MSC line HS-5, which was used in monoculture (HS-5) and cocultured with normal peripheral blood mononuclear cells (PBMCs) and AML cell lines (MOLM-13 and HL-60) for 5 days, and then induced AML-associated MSCs (iAML-MSC) were sorted (CD90+) using Fluorescence Activated Cell Sorting (FACS) (Appendix A). Following this coculture period, we performed a detailed analysis of the metabolic characteristics of the sorted MSCs, identified as HS-5, HS-5 + PBMCs, and HS-5 + MOLM-13/H60 cells (iAML-MSC). The results demonstrated a significant alteration in the metabolic profile of the iAML-MSC. Specifically, we observed a marked reduction in both OXPHOS at the basal and maximum level (Figure 3A–C) and glycolytic activity (Figure 3D–F) in the cocultured MSCs when compared to control MSCs that were not exposed to AML cells. This metabolic suppression was further supported by a decrease in reactive oxygen species (ROS) levels (Figure 3G) and a reduction in mitochondrial mass (Figure 3H) in iAML-MSCs. Furthermore, HS-5 cells cocultured with AML cells exhibit reduced proliferative capacity, indicating that iAML-MSCs undergo a state of senescence (Figure 3I). These findings are consistent with previous studies conducted on murine models, where similar metabolic alterations were noted in MSCs cocultured with AML cells. The convergence of results across both human and murine systems underscores the potential for AML cells to significantly influence the metabolic capacity of MSCs, suggesting a broader implication for the interaction between cancer cells and the surrounding stromal environment. In addition, we added healthy PBMCs for 3 days after removing the AML cells from a 5-day coculture with HS-5 cells in order to assess whether MSCs previously in contact with AML cells could be functionally rescued. The findings indicate that exposure to healthy PBMCs did not reverse the leukemia-supportive phenotype of AML-educated MSCs (Appendix A). In summary, our data indicate that human AML cells can induce a profound metabolic reprogramming in MSCs, characterized by diminished OXPHOS and glycolytic pathways, alongside reductions in ROS production and mitochondrial content. This metabolic shift may have important implications for the role of MSCs in the tumor microenvironment and their interactions with malignant cells.

### 2.4. Metabolic Capacity Assessment in Immortalized Human AML-MSCs

To further substantiate our findings in a non-culture-dependent context, we employed immortalized human MSCs derived from the bone marrow of both healthy donors (referred to as HD-MSCs) and AML patients (designated as AML-MSCs) (Appendix A) [20]. Phenotypic characterization of these immortalized MSCs confirmed the expression of established MSC surface markers, including CD73, CD90, and CD105. Furthermore, these cells demonstrated the capacity to differentiate into adipocytes, chondrocytes, and osteocytes, affirming their multipotent nature [21,22]. Subsequently, we assessed the metabolic profiles of HD-MSCs and AML-MSCs utilizing seahorse mito-stress and glycostress assays. The results revealed that AML-MSCs exhibited significantly reduced rates of OXPHOS (Figure 4A,B) and glycolysis (Figure 4C,D) compared to their HD-MSC counterparts. Additionally, flow cytometry analysis indicated that AML-MSCs possessed markedly lower levels of ROS (Figure 4E) and mitochondrial mass (Figure 4F) relative to HD-MSCs, but less reduced membrane potential (*p* = 0.0569) (Figure 4G). In addition, AML-MSCs displayed a more pronounced reduction in proliferative capacity compared to HD-MSCs (Figure 4H). In conclusion, our comprehensive analysis of both murine and human AML-MSC models elucidates significant metabolic disparities between control MSCs and those influenced by leukemic conditions. These findings not only enhance our understanding of the metabolic reprogramming induced by AML but also suggest potential therapeutic strategies aimed at mitigating the adverse effects of leukemia on MSC function.

## 3. Discussion

The prognosis remains poor for the majority of adult AML patients. Only 25% of patients survive more than five years after the initial diagnosis [23]. The proliferation of AML cells depends on intrinsic genetic and epigenetic factors and interaction with MSCs, an essential element of the BM hematopoietic niche. The presence of AML cells altered MSC function since the MSCs failed to support the function of healthy hematopoietic stem cells (HSCs) in myeloid malignancies [24,25]. Kim et al. reported that an altered stroma function could predict the prognosis of leukemic patients [6].

Our study reveals significant metabolic adaptations in MSCs influenced by acute myeloid leukemia (AML) cells, highlighting a unique metabolic vulnerability within the leukemia-associated stromal microenvironment. The results indicate that AML cells impose profound metabolic restrictions on MSCs, which may directly contribute to AML progression and chemoresistance. Specifically, we observed significant downregulation of OXPHOS and glycolysis in MSCs exposed to AML, corroborated by reduced mitochondrial DNA content, oxygen consumption rates (OCRs), extracellular acidification rates (ECARs), glucose consumption, and lactate production. These findings provide new insights into the metabolic interactions between leukemic cells and their supporting stroma, suggesting that AML cells can effectively “reprogram” MSCs to create a microenvironment supporting leukemia survival and resistance to treatment [6,13,25]. The reduction in proliferative capacity and the decrease in OXPHOS and glycolysis in AML-MSCs suggest a metabolic shift favoring AML cells’ survival by reserving key nutrients and bioenergetic pathways for leukemic cells. This aligns with previous studies showing that leukemic cells actively acquire functional mitochondria from MSCs, which may provide them with additional metabolic flexibility and resources for sustaining their high energy demands [13,14]. By imposing metabolic suppression on MSCs, AML cells may conserve glucose, oxygen, and other essential resources, enhancing their own proliferation and survival while making MSCs less capable of supporting normal hematopoiesis. This reprogramming could represent a survival strategy for AML, potentially contributing to the “niche dominance” of leukemic cells over healthy progenitors within the bone marrow. Furthermore, the observed reduction in ROS and mitochondrial mass in AML-MSCs aligns with previous reports that diminished mitochondrial function in stromal cells can contribute to an environment that supports AML chemoresistance [10]. This metabolic shift may enable AML cells to evade chemotherapy by reducing the reliance on mitochondrial function, which is often a target of many chemotherapeutic agents.

Previous studies have demonstrated that activation of Notch signaling suppresses the transcription of genes involved in glycolysis and mitochondrial complex I activity, leading to decreased mitochondrial respiration, reduced reactive oxygen species production, and diminished AMPK signaling [26]. These findings suggest that Notch signaling may play a critical role in regulating cellular metabolism. In line with this, our prior work established that Notch signaling is significantly upregulated in AML-associated mesenchymal stromal cells (AML-MSCs), as observed in both murine models and human transcriptomic datasets [10]. Functional studies further revealed that enforced Notch activation in MSCs promotes AML cell proliferation, indicating a supportive role in leukemic progression. Notably, murine MSCs engineered to express the constitutively active intracellular domain of Notch1 (MS5-ICN) exhibited marked reductions in both oxidative phosphorylation (OXPHOS) and glycolytic activity when compared to empty vector controls (MS5-EV). These results support the hypothesis that aberrant Notch signaling contributes to the metabolic reprogramming of MSCs in the leukemic microenvironment, potentially creating a niche that favors AML cell survival and expansion. 

Moreover, the interplay between metabolic suppression in AML-MSCs and immune modulation within the bone marrow niche warrants further exploration. The reduced metabolic activity of AML-MSCs may also affect immune cell functions, potentially creating an immunosuppressive microenvironment that further supports AML progression. Moreover, the interplay between metabolic suppression in AML-MSCs and immune modulation within the bone marrow niche warrants further exploration. The reduced metabolic activity of AML-MSCs may also affect immune cell functions, potentially creating an immunosuppressive microenvironment that further supports AML progression. Recently, Ikeda et al. reported that cancer cells can transmit mtDNA mutations to tumor-infiltrating lymphocytes (TILs), leading to mitochondrial dysfunction [27,28].

The metabolic adaptations in MSCs influenced by AML cells represent a novel target for therapeutic intervention. Targeting these metabolic pathways in AML and MSCs’ interaction by blocking mitochondria transferring [12,28,29,30] could disrupt the supportive role of the stromal environment, reducing leukemic cell proliferation and potentially increasing their sensitivity to treatment. In particular, agents that restore or enhance mitochondrial function in MSCs, or inhibit mitochondrial transfer [31,32] to AML cells could limit AML progression. Additionally, therapeutic strategies aimed at maintaining or boosting OXPHOS and glycolysis in MSCs may restore normal hematopoietic support and reduce leukemic niche formation, creating an environment less favorable to AML persistence.

Strategies should aim to restore MSC function rather than exacerbate metabolic impairment. Such agents could be used in conjunction with existing chemotherapies to increase AML cell susceptibility by disrupting the nutrient and energy support from MSCs while minimizing adverse effects on the stromal compartment. Furthermore, mitochondrial transfer inhibitors represent a promising area of study; limiting this process could hinder the metabolic reprogramming AML imposes on MSCs and reduce leukemic cell adaptability, potentially without further compromising MSC viability [29].

Understanding the crosstalk between metabolic alterations and immune evasion [32] could lead to combined metabolic and immunotherapeutic approaches to target AML more effectively.

While our study offers valuable insights, several limitations should be addressed in future research. First, the metabolic responses observed in murine and human MSCs may differ slightly due to species variations. Further studies in human primary MSCs from AML patients are necessary to confirm the translatability of our findings. Additionally, the exact molecular mechanisms by which AML cells induce metabolic suppression in MSCs remain unclear. Future research should investigate the signaling pathways and intercellular communication mediators responsible for this metabolic reprogramming, such as exosomes, cytokines, and other soluble factors released by AML cells.

In summary, our study underscores the metabolic vulnerability of MSCs in the AML microenvironment, revealing a shift towards suppressed OXPHOS and glycolysis that appears to favor leukemic cell survival and chemoresistance. By elucidating these metabolic dependencies, we open the door to novel therapeutic strategies aimed at restoring metabolic balance within the leukemia niche. Targeting the metabolic adaptations of MSCs offers a promising approach to disrupt AML progression and enhance treatment responses, potentially improving outcomes for patients with this challenging malignancy.

## 4. Materials and Methods

### 4.1. Mouse Models and Transplantation

Wild-type C57BL/6J mice were used to establish the MLL-AF9-induced leukemic model, as described previously [33,34]. Briefly, total bone marrow (BM) cells were isolated and lineage-depleted cells were enriched using MACS columns (-130-042-401 and 130-090-858, miltenyi biotec, Bergisch Gladbach, Germany) according to the manufacturer’s protocol. These lineage-negative cells were then transduced with a retrovirus encoding the MLL-AF9 oncogene (MSCV-MLL-AF9-IRES-GFP), which includes a GFP reporter for tracking transduction efficiency. Following transduction, GFP^+^ cells were sorted via flow cytometry and cultured for 2 days in a stem cell medium (SCM-IMDM supplemented with 20% FBS, 1% P/S, 10 ng/mL IL-3, 10 ng/mL IL-6 and 20 ng/mL mSCF). The cultured cells were then transplanted into lethally irradiated C57BL/6J recipient mice to induce leukemia. Once the mice developed leukemia, they were euthanized, and bone marrow cells were harvested. Bones were crushed and digested with collagenase to facilitate cell recovery. The isolated cells were seeded in IMDM medium supplemented with 20% FBS and cultured for 2 weeks. Mesenchymal stromal cells (MSCs) were identified as adherent cells and characterized by flow cytometry using MSC surface markers (List of markers). All experiments were performed using MSC populations with ≥95% purity. All mice were bred and maintained under specific pathogen-free conditions. 

### 4.2. AML Cell Lines

The human acute myeloid leukemia (AML) cell lines THP-1 and MOLM-13 were obtained from the American Type Culture Collection (ATCC, Manassas, VR, USA) and cultured in RPMI-1640 medium (Sigma-Aldrich, Hamburg, Germany) supplemented with 20% fetal bovine serum (FBS, PAN^TM^ Biotech, Aidenbach, Germany) and 1% penicillin/streptomycin (Sigma-Aldrich). The murine AML cell line C1498, retrovirally transduced with the pLEGFP plasmid (C1498-GFP), was kindly provided by Dr. Justin Kline (University of Chicago, Chicago, IL, USA). C1498 cells were maintained in RPMI-1640 medium (Sigma-Aldrich, Darmstadt, Germany) with 10% FBS (PAN^TM^ Biotech) and 1% penicillin/streptomycin (Gibco, Life Technologies, Darmstadt, Germany) [19].

### 4.3. Microarray Analysis

To investigate gene expression changes in the bone marrow microenvironment during leukemia progression, we transplanted primary MLL-AF9-induced leukemic cells into sub-lethally irradiated (3 Gy) C57BL/6 (BL6) recipient mice. As a control, wild-type, non-leukemic bone marrow cells were transplanted into identically irradiated BL6 mice. Upon confirmation of leukemia development in the experimental group, the mice were sacrificed, and MSCs (CD45-Lin-CD31-Sca-1 + CD51+) were isolated using a BD Biosciences FACS Vantage SE flow cytometer equipped with BD FACSDiva v9.0 Software (Heidelberg, Germany), as detailed in Section 4.1. Control mice were processed simultaneously to ensure experimental consistency. Total RNA was extracted from sorted MSCs, and gene expression profiling was performed using microarray analysis, as shown previously [16,17,18]. The raw microarray data are publicly available through the Gene Expression Omnibus (GEO) under accession number GSE102928.

### 4.4. RNA Sequencing

To perform RNA sequencing (RNA-seq) analysis, bone marrow cells were isolated from C57BL/6J (BL6) mice and expanded for mesenchymal stem cells (MSCs), as described in Section 4.1. The resulting MSCs were cocultured either with MLL-AF9-induced leukemic bone marrow cells (leukemic condition) or with lineage-negative (Lin^−^) non-leukemic cells (non-leukemic condition) for four weeks. MSCs cultured alone without coculture served as the control. Following the coculture period, MSCs (*CD45–* Sca1+ CD51+) were isolated via fluorescence-activated cell sorting (FACS), and total RNA was extracted using the RNeasy Plus Micro Kit (Qiagen, Hilden, Germany), according to the manufacturer’s instructions.

Transcriptome profiling was performed using cDNA libraries prepared with the TruSeq Stranded mRNA Sample Preparation Kit (Illumina, San Diego, CA, USA). Sequencing was carried out on an Illumina HiSeq 2500 platform. Quality control of the raw reads was conducted using FastQC (v0.11.7; Babraham Bioinformatics, Cambridge, UK). Transcript abundance was estimated using Kallisto (v0.43.0), aligned to the GRCm38 mouse reference genome. Differential gene expression analysis was conducted using the Sleuth (v0.30.0) package in R, comparing MSCs cultured alone with those cocultured with leukemic or non-leukemic cells. Multiple testing correction was applied using the Benjamini–Hochberg procedure, with a false discovery rate (FDR) threshold of <1%.

Gene Set Enrichment Analysis (GSEA) was performed using the fgsea R package (v1.10.0), employing hallmark gene sets from the Molecular Signatures Database (MSigDB) and an FDR threshold of 5%.

### 4.5. Coculture Experiments with MSCs and AML Cells

Wild-type mice were sacrificed, and BM cells were collected by flushing femurs and tibiae with FACS buffer (DPBS (Gibco, Life Technologies, Darmstadt, Germany) containing 2.5% fetal bovine serum (FBS) (PANT Biotech and 1% penicillin/streptomycin (Gibco)). The endosteal stromal cells were collected as previously described with a minor modification [35]. In brief, flushed bones (femurs, tibia, and humeri) were crushed by pestle and mortar, washed with HBSS with 2.5% FBS and 1% penicillin streptomycin until the small bone chips became white. The endosteal stromal cells were then extracted by digestion with 3 mg/mL collagenase 1 (Worthington, Troisdorf, Germany). Red blood cells were removed by incubating with ammonium chloride lysing reagent (BD Biosciences, Heidelberg, Germany) for 7 min. Thereafter, the mononuclear cells were cultured in the adherent flask with Iscove’s Modified Dullbecco medium (IMDM) (Gibco, Life Technologies, Darmstadt, Germany) containing 20% fetal bovine serum (FBS) (PANT Biotech, Aidenbach, Bayern, Germany) and 1% penicillin/streptomycin (Gibco) and incubated at 37 °C in a humidified atmosphere containing 5% CO_2_. Half of the medium was replaced twice a week. When the MSCs were 80–90% confluent, they were washed by DPBS (Gibco) and then split by trypsin-EDTA (Gibco) and replated into new adherent flask. After splitting for 2 or 3 times, the MSCs were cultured alone or with AML cells and normal lineage-negative cells with 1:5 for 7 days. After that, the MSCs were sorted as CD45- CD51+ Sca-1+ for analysis. Primary AML and Lin- cells were cultured in IMDM medium containing 20% FBS, IL-3, IL-6, SCF, and 1% penicillin–streptomycin for six days.

### 4.6. Glucose and Lactate Measurements

Glucose and lactate levels were measured on days 3 and 5 from the culture supernatants collected after sorting MSCs (CD45^−^ CD51^+^ Sca-1^+^). Samples were submitted to the diagnostic laboratory of Prof. Dr. med. Jerzy-Roch Nofer at the University Hospital Münster for biochemical analysis. Glucose consumption and lactate secretion were determined by comparing the measured values to those of the corresponding cell-free (empty) media controls. All values were normalized to the number of MSCs present at the time of sample collection.

### 4.7. Mitochondrial DNA (mtDNA) Copy Number

The mitochondrial DNA copy number was assessed using quantitative PCR of mtDNA and nuclear DNA (nuDNA), as previously described [36,37,38]. Primer sequences are listed in Appendix A.

### 4.8. Seahorse Metabolic Assays

Seahorse metabolic flux assays were performed as previously described using Agilent XFe96 and XFp analyzers [36,37,38]. Briefly, XF sensor cartridges were hydrated overnight in a non-CO_2_ incubator, and assay plates were pre-coated with Poly-D-Lysine (PDL) to enhance cell adherence. Cells were seeded at a density of 20,000 cells per well in XF Base Medium supplemented with glucose, glutamine, and sodium pyruvate, as detailed in Appendix A.

For the mito-stress test, after baseline oxygen consumption rate (OCR) measurements, cells were sequentially treated with oligomycin, FCCP, and a mixture of rotenone and antimycin A to assess mitochondrial respiration parameters. For the glycolysis stress test, the extracellular acidification rate (ECAR) was measured under basal conditions and following sequential injections of glucose, oligomycin, and 2-deoxyglucose (2-DG). At the end of each assay, cells were stained with Hoechst 33342 for normalization to cell number.

### 4.9. Flow Cytometry

Mitochondrial mass (mito mass), reactive oxygen species (ROS) and mitochondrial membrane potential (MMP) were measured as described previously [36,37,38]. For mitochondrial membrane potential (MMP) and reactive oxygen species (ROS) detection, 0.5 × 10^5^ cells were resuspended in cell culture media prewarmed to 37 °C and stained with 50 nM tetramethylrhodamine methyl ester perchlorate (TMRE) (ab113852, Abcam, Cambridge, UK) or 5 μM cell rox deep red (C10422, ThermoFischer, Waltham, MA, USA), followed by incubation for 30 min at 37 °C in a CO_2_ incubator and measured directly by flow cytometry. Carbonyl cyanide-p-trifluoromethoxyphenylhydrazone (FCCP) (ab120081, Abcam) was used as a negative control for TMRE staining. Mitochondrial mass was detected by resuspending 0.5 × 10^5^ cells with cell culture media prewarmed to 37 °C without FCS, followed by staining with 50 nM mitotracker deep red (M22426, ThermoFischer), incubated for 20 min at 37°C in a CO_2_ incubator and measured by FACS.

### 4.10. Apoptosis Assay

Apoptosis was assessed using the Annexin V-FITC/Propidium Iodide (PI) staining kit (e.g., BioLegend, Thermo Fisher) following the manufacturer’s instructions. Briefly, cells were harvested, washed twice with cold PBS, and resuspended in 100 µL of binding buffer at a concentration of 1 × 10^6^ cells/mL. Annexin V-FITC (5 µL) and PI (5 µL) were added to each sample and incubated for 15 min at room temperature in the dark. After incubation, 400 µL of binding buffer was added, and the samples were analyzed immediately by flow cytometry (e.g., BD FACSCanto II). Data were analyzed using FlowJo software v10.9.0.

### 4.11. Cell Proliferation

Cell proliferation was assessed by manual cell counting using the Trypan Blue exclusion method. Viable (unstained) and non-viable (blue-stained) cells were counted using a hemocytometer, and total live cell numbers were used to evaluate proliferation over time.

### 4.12. Statistical Analysis

Statistical differences were evaluated using Student’s *t*-test and one-way ANOVA analysis using GraphPad software version 6. A *p*-value of <0.05 was considered statistically significant.

## Figures and Tables

**Figure 1 ijms-26-08301-f001:**
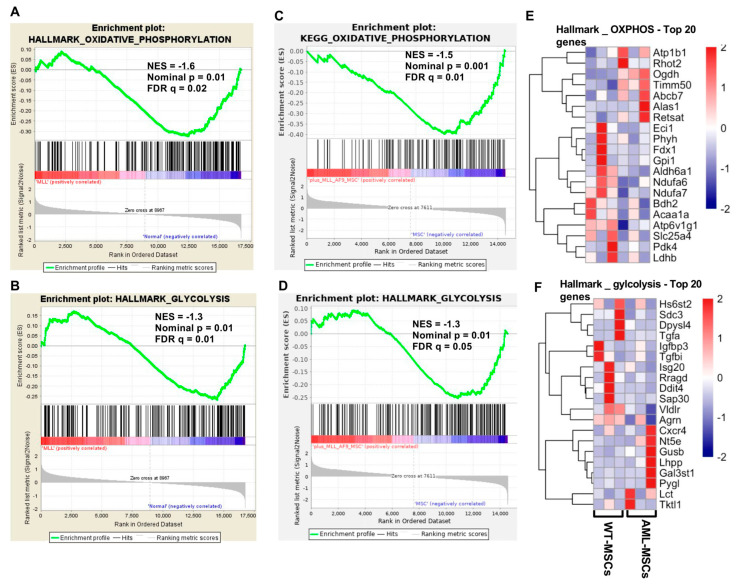
Molecular changes in MSCs cocultured with AML cells and in MSCs from leukemic and non-leukemic mice. (**A**,**B**) Microarray data showed enrichment plots of GSEA for CD45-Lin-CD31-Sca-1 + CD51 + AML-MSCs, normalized enrichment score (NES) = −1.60 and *p*-value = 0.01 for oxidative phosphorylation, (NES) = −1.3 and *p* value = 0.01 for glycolysis signaling. (**C**,**D**) Schematic illustration of the experimental setting of MSCs cocultured alone or with MLL-AF9 and Lin- cells for 4 weeks; RNA sequence data showed enrichment plots of GSEA for MSCs cocultured with MLL-AF9 cells (*n* = 3) compared to MSCs alone as control (*n* = 3). (**E**) Scaled expression values of top 20 genes in oxidative phosphorylation. (**F**) Scaled expression values of top 20 genes in glycolysis.

**Figure 2 ijms-26-08301-f002:**
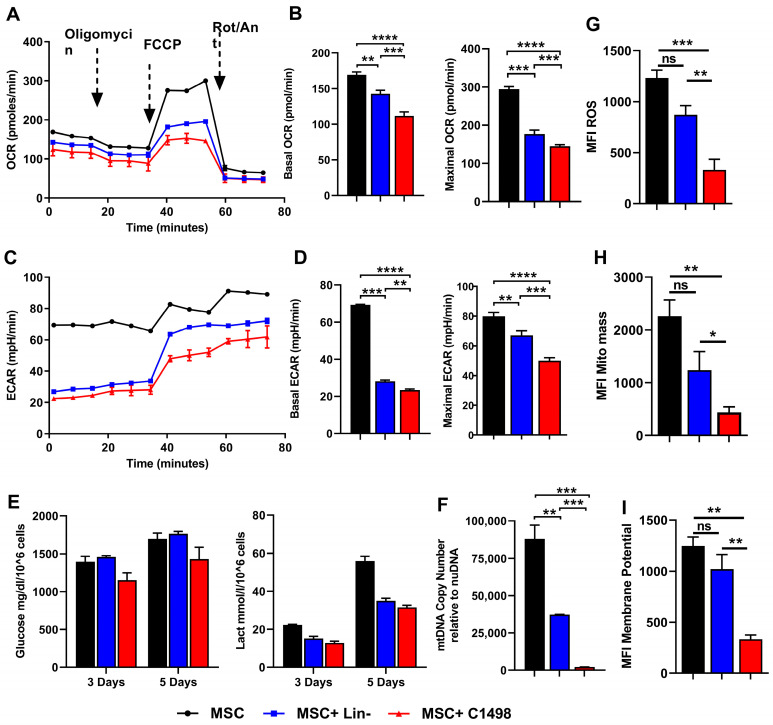
AML cells cocultured with MSCs alter the cell metabolism of MSC (**A**) Murine MSCs were cocultured with C1498 cells for 5 days, and then sorted CD45-CD51 + SCA-1 + MSCs were subjected to Seahorse analyses to monitor OCR as an assessment of oxidative phosphorylation. (**B**) OCR at basal level (left panel) and at maximal level (right panel) (**C**) Seahorse analyses to monitor ECAR as an assessment of glycolysis. (**D**) ECAR at basal level (left panel) and at maximal level (right panel) (**E**) Sorted MSCs were cultured for 3 and 5 days to measure the glucose consumed (left panel) and lactate secreted (right panel) by MSCs. (**F**) The mitochondrial DNA (mtDNA) copy number was calculated by real-time amplification of mtDNA and nuDNA genes, as reported. The primers used in mtDNA measurement are listed in Appendix A. (**G**) Flow cytometry analysis shows a significant reduction in ROS production in AML-MSCs. (**H**) Flow cytometry analysis shows a significant reduction in mitochondrial mass (mito mass) in AML-MSCs. (**I**) Flow cytometry analysis shows a smaller reduction in membrane potential (TMRE) in AML-MSCs. The data represent of the three independent experiments. A one-way ANOVA analysis was used to determine the significance, * *p* ≤ 0.05, ** *p* ≤ 0.01 and *** *p* ≤ 0.005 and **** *p* ≤ 0.001.

**Figure 3 ijms-26-08301-f003:**
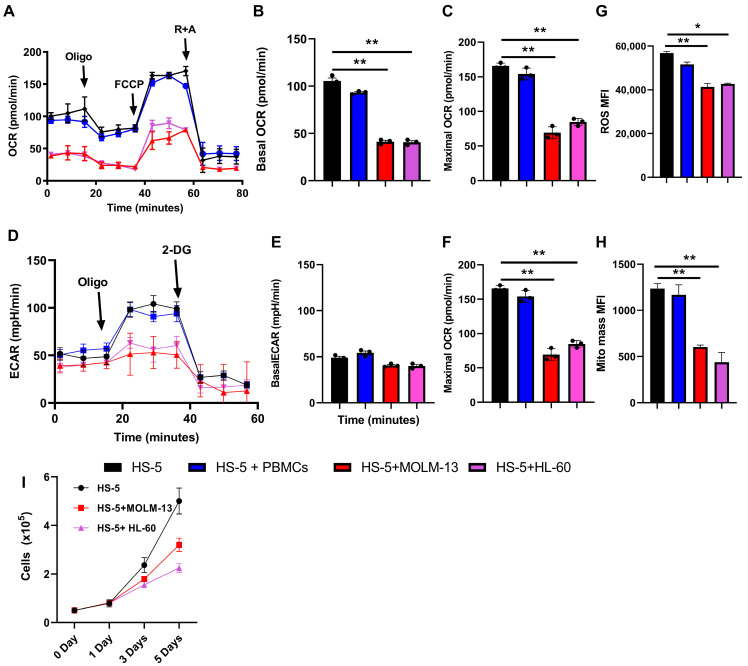
AML cells cocultured with HD-MSCs alter their metabolism. (**A**) Human healthy donor MSCs (HS-5) cocultured either alone or with PBMCs, and MOLM-13 and HL-60 cells (iAML-MSCs) for 5 days; then, FACS-sorted CD90+ MSCs were subjected to Seahorse analyses to monitor OCR as an assessment of oxidative phosphorylation. (**B**,**C**) iAML-MSC shows a significant reduction in the basal OCR level and maximal level. (**D**) Seahorse glycostress assay to monitor the rate of glycolysis. (**E**,**F**) iAML-MSCs showed a significant reduction in maximum ECAR. (**G**) FACS analysis showed a significant reduction in ROS levels in iAML-MSCs. (**H**) FACS analysis showed a significant reduction in mitochondrial mass in iAML-MSCs. (**I**) HS-5 cocultured with MOLM-13 and HL-60 cells (iAML-MSC) displays reduced proliferative capacity compared to monoculture. The data represent the three independent experiments. A one-way ANOVA analysis was used to determine the significance, * *p* ≤ 0.05, ** *p* ≤ 0.005.

**Figure 4 ijms-26-08301-f004:**
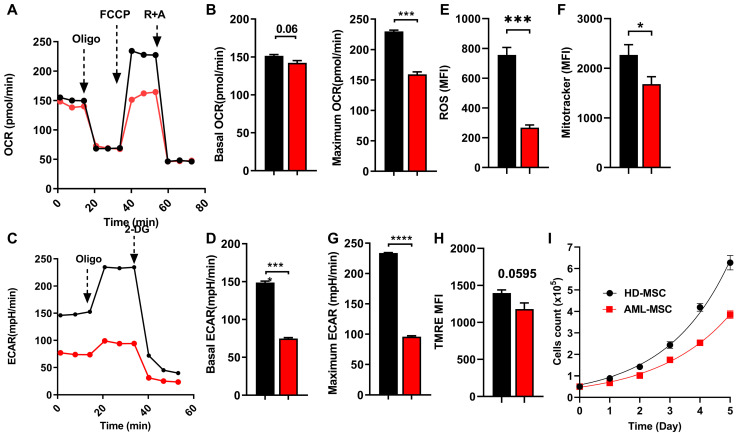
Metabolism alterations in human AML-MSCs vs. HD-MSCs. (**A**) Human HD-MSCs and AML-MSCs subjected to Seahorse mito-stress to monitor OCR as an assessment of oxidative phosphorylation. (**B**) AML-MSCs showed a significant reduction in basal OCR (left panel) and maximum OCR (right panel). (**C**) Seahorse glycotress assay to monitor ECAR as an assessment of glycolysis. (**D**) AML-MSCs showed a significant reduction in basal ECAR (left panel) and maximum ECAR (right panel). (**E**) Flow cytometry analysis shows a significant reduction in ROS production in AML-MSCs. (**F**) Flow cytometry analysis shows a significant reduction in mitochondrial mass (mito mass) in AML-MSCs. (**G**) Flow cytometry analysis shows a reduction in membrane potential (TMRE) in AML-MSCs. (**H**) AML-MSCs display reduced proliferative capacity compared to HD-MSCs. **(I)** AML-MSCs display reduced proliferative capacity compared to HD-MSCs. The data represent a single experiment of the three independent experiments. Student’s *t*-test was used to determine the significance, * *p* ≤ 0.05, *** *p* ≤ 0.005 and **** *p* ≤ 0.001.

## Data Availability

The raw data supporting the conclusions of this article will be made available by the authors on request.

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
