# Peer review of "Impact of Acute Myeloid Leukemia Cells on the Metabolic Function of Bone Marrow Mesenchymal Stem Cells"

_ijms, 2025, doi:10.3390/ijms26178301_

Round 1

Reviewer 1 Report (Previous Reviewer 2)

Comments and Suggestions for Authors

The authors have sufficiently addressed my previous queries.

Author Response

Referee #1 (Comments to the Author):

In the manuscript “Impact of Acute Myeloid Leukemia Cells on the Metabolic Function of Bone Marrow Mesenchymal Stem Cells”, Ahmed et al. studied transcriptional and metabolic alterations in bone marrow mesenchymal stem cells (BMSCs) derived from AML in both mouse and human. The manuscript is well-structured and contains thoughtfully presented information. I have some comments.

Comments:

  1. Please check that the figure numbers in the manuscript are correct. I think that lines 113-114 and 116 are Figure 2...

Response: Thank you for the reviewer’s valuable comment. I have corrected the figure number (in blue color).

  1. Line 132, Figure 2 legend. What do the words in bold mean?

Response: Thank you for the reviewer’s insightful comment. I have corrected it (in supplemental table 2).

  1. Section 2.3. It would be useful to have information on whether there are any differences between iAML-MSCs and HS-5 on flow cytometry other than CD45-CD90+, and whether there are any differences in the rate of proliferation when continued in culture.

Response: Thank you for the reviewer’s valuable comment. In our previous publication (Ahmed et al., 2021, BJH), we reported that there are no differences in immunophenotypes between AML-MSCs and HD-MSCs. However, AML-MSCs exhibit lower proliferation rates compared to HD-MSCs.

  1. Line 139. FACs? It is probably better to explain abbreviations by first spelling them out in full.

Response: Thank you for the reviewer’s comment. I have spelled them.

  1. Line 172-175. Even if the antigens that are positive for the flow patterns in HD-MSCs and AML-MSCs are the same, are the positive frequencies the same? Please explain in detail.

Response: Thank you for reviewer comments. In our previous publication (Ahmed et al., 2021, BJH), we reported that there are no differences in immunophenotypes between AML-MSCs and HD-MSCs.

  1. Line 253-267. The authors also mention the possibility that AML-MSCs may affect the function of immune cells. Is it possible that the abnormal mitochondrial DNA in AML cells is harming the MSCs and surrounding immune cells? Recently, Nature published some interesting data (Ikeda et al. Nature. 2025 Jan 22. doi: 10.1038/s41586-024-08439-0. Online ahead of print). If you think it might be relevant to the theme of the manuscript, why not add it to the discussion?

Response: Thank you for the reviewer’s valuable comment. This is indeed an interesting article that highlights how cancer cells influence microenvironment metabolism to evade immune responses. I have added a sentence addressing this point in the discussion section.

Reviewer 2 Report (New Reviewer)

Comments and Suggestions for Authors

The manuscript "Impact of acute myeloid leukemia cells on the metabolic function of bone marrow mesenchymal stem cells" presents interesting information, although I do not consider the authors' claims to be fully supported by the experimental results. Despite presenting modifications suggested by some other reviewers, I do not consider the improvements sufficient for publication. Also, the manuscript is quite confusing so I suggest improving the fluidity and organization.

  • In your statistical analysis section, you don't mention how you determined your data were parametric. Furthermore, in Figure 2, you have three groups, so using Student's t-test is inappropriate, and appropriate corrections should be made.
  • Your figure 1A is incomplete, it only shows what you do with the leukemic cells but it is missing what happens with the control cells.
  • What does "Figure. 1 Band 1D" mean?
  • There is no reference to Figure 1C in the text.
  • If OXPHOS and glycolytic activity are reduced in MSCs, what metabolic pathway do they use to survive? This generally leads to cell death. Apoptosis and senescence studies are necessary
  • For studies with flow cytometry, it is advisable to add histograms or scatter plots, at least in supplementary figures.
  • How do reprogrammed MSCs (with low OXPHOS and glycolysis) create a microenvironment that favors leukemia survival and treatment resistance?
  • How could OXPHOS inhibitors or glycolysis activators in MSC or AML-MSCs (both with low glycolysis and OXPHOS) help in the treatment?
  • The materials and methods section should be more descriptive so that it is reproducible and not leave most of the details to other manuscripts.

Author Response

Referee #2 (Comments to the Author):

The manuscript "Impact of acute myeloid leukemia cells on the metabolic function of bone marrow mesenchymal stem cells" presents interesting information, although I do not consider the authors' claims to be fully supported by the experimental results. Despite presenting modifications suggested by some other reviewers, I do not consider the improvements sufficient for publication. Also, the manuscript is quite confusing so I suggest improving the fluidity and organization.

In your statistical analysis section, you don't mention how you determined your data were parametric. Furthermore, in Figure 2, you have three groups, so using Student's t-test is inappropriate, and appropriate corrections should be made.

Response: Thank you for this important observation. We acknowledge the omission in our statistical analysis section. We have now updated the Methods section to specify that we assessed the distribution of our data using the Shapiro–Wilk test for normality and Levene’s/Brown–Forsythe test for homogeneity of variances. For Figure 2, since the dataset included three groups, we agree that using multiple Student’s t-tests is not appropriate due to the increased risk of Type I error.

We have corrected this by performing a one-way ANOVA to test for overall differences among the groups. Given that variances were found to be significantly different (as confirmed by the Brown–Forsythe test), we followed up with the Welch’s ANOVA, which does not assume equal variances. Additionally, we performed Games–Howell post-hoc tests for pairwise comparisons between groups.

These changes are now clearly described in the revised manuscript in both the Statistical Analysis subsection and the Figure 2and 3legends. The results and statistical significance levels remain consistent with our original conclusions.

Your figure 1A is incomplete, it only shows what you do with the leukemic cells but it is missing what happens with the control cells.

Response: We thank the reviewer for this valuable comment. Indeed, Figure 1A currently depicts only the experimental procedure for leukemic cells and does not clearly illustrate the control conditions. To address this, we have revised Figure 1A to explicitly include both experimental groups: MSCs sorted from leukemic mice transplanted with MLL-AF9 cells (AML-MSCs) as well as MSCs sorted from non-leukemic control mice transplanted with normal bone marrow cells (WT-MSCs). This updated figure now clearly illustrates both experimental workflows (supp. figure 1).

What does "Figure. 1 Band 1D" mean?

Response: We thank the reviewer for this comment. Its corrected.

 (Page number 3 and lines 100-120).

There is no reference to Figure 1C in the text.

Response: We thank the reviewer for this comment. Its corrected.

(Page number 3 and lines 100-120).

If OXPHOS and glycolytic activity are reduced in MSCs, what metabolic pathway do they use to survive? This generally leads to cell death. Apoptosis and senescence studies are necessary

Response: We appreciate the reviewer’s insightful comment. In our study, we demonstrate that AML-MSCs exhibit reduced proliferative capacity in both murine and human models, as shown in Supplementary Figure 1I, Figure 3G, and Figure 4H. In line with the reviewer’s concern, we agree that reduced oxidative phosphorylation (OXPHOS) and glycolytic activity would typically compromise cell viability. We thank the reviewer for prompting this important clarification.

For studies with flow cytometry, it is advisable to add histograms or scatter plots, at least in supplementary figures.

Response: We thank the reviewer for this comment. We updated the accordingly in supplement figures.

How do reprogrammed MSCs (with low OXPHOS and glycolysis) create a microenvironment that favors leukemia survival and treatment resistance?
Response: Thank you for this important question. We have expanded the discussion section in the revised manuscript to clarify this point more thoroughly.

Mesenchymal stromal cells (MSCs) that are reprogrammed into a metabolically quiescent state characterized by reduced oxidative phosphorylation (OXPHOS) and glycolysis play a critical role in shaping a leukemogenic bone marrow microenvironment through multiple mechanisms.

These metabolically inactive MSCs mimic the hypoxic endosteal niche, which supports leukemic stem cell (LSC) dormancy, chemoresistance, and long-term survival by maintaining low levels of reactive oxygen species (ROS) and limiting metabolic activity (1, 2). In addition, such MSCs secrete factors including CXCL12, IL-6, and TGF-β, which enhance leukemia cell survival, preserve stemness, and suppress apoptosis (3).

Notably, despite their own reduced metabolic output, MSCs can support leukemic bioenergetics by transferring mitochondria via tunneling nanotubes, thereby enhancing OXPHOS capacity and promoting survival of leukemia cells under metabolic stress (4, 5).

Moreover, quiescent MSCs help establish an immunosuppressive microenvironment by producing indoleamine 2,3-dioxygenase (IDO), expressing PD-L1, and promoting regulatory T cell (Treg) function mechanisms that collectively facilitate immune evasion by leukemia cells (6).

Together, these mechanisms underscore our conclusion that metabolically reprogrammed MSCs foster a specialized niche that promotes leukemia persistence and resistance to therapy. We have incorporated this explanation in the revised discussion (Page, Lines 270 – 300)

How could OXPHOS inhibitors or glycolysis activators in MSC or AML-MSCs (both with low glycolysis and OXPHOS) help in the treatment?
Response: We thank the reviewer for this insightful question. Although AML-educated MSCs exhibit a suppressed metabolic profile characterized by low OXPHOS and glycolysis, recent studies suggest that their residual metabolic activity especially mitochondrial transfer and metabolic plasticity can still play a critical role in leukemic cell survival and treatment resistance.

OXPHOS inhibition in MSCs may have therapeutic value not by directly affecting the MSCs' metabolism, but by preventing mitochondrial transfer from MSCs to leukemic cells. It has been shown that MSCs can donate mitochondria to AML blasts, thereby enhancing leukemic oxidative phosphorylation and contributing to drug resistance (4, 5).  The combination therapies that target both leukemic cells and their supportive niche, such as OXPHOS inhibitors with BCL-2 inhibitors like venetoclax, have shown synergistic anti-leukemic effects(7). Another study demonstrated that Metformin and the combination of venetoclax and daratumumab significantly slowed tumor progression and reduced leukemic burden both in vitro and in vivo, primarily through the inhibition of OXPHOS and the suppression of mitochondrial transfer from MSCs to acute myeloid leukemia (AML) cells (8, 9). We have now clarified this point in the revised manuscript, discussion (Page, Lines 300 – 319)

The materials and methods section should be more descriptive so that it is reproducible and not leave most of the details to other manuscripts.

Response: Thank you for this valuable comment. In response, we have substantially revised and expanded the Materials and Methods section to ensure greater clarity and reproducibility. We have now included detailed descriptions of the experimental procedures, cell culture conditions, treatment protocols, and analytical methods. Wherever applicable, we have also specified reagent sources, concentrations, incubation times, and data analysis approaches to minimize ambiguity. These updates aim to provide a comprehensive and self-contained account of our methods, rather than relying on references to prior manuscripts.

Please see the revised Materials and Methods section (Pages 9–11) for these additions.

  1. Saito Y, Uchida N, Tanaka S, Suzuki N, Tomizawa-Murasawa M, Sone A, et al. Induction of cell cycle entry eliminates human leukemia stem cells in a mouse model of AML. Nat Biotechnol. 2010;28(3):275-80.
  2. Méndez-Ferrer S, Bonnet D, Steensma DP, Hasserjian RP, Ghobrial IM, Gribben JG, et al. Bone marrow niches in haematological malignancies. Nat Rev Cancer. 2020;20(5):285-98.
  3. Liu Y, Cao X. Characteristics and Significance of the Pre-metastatic Niche. Cancer Cell. 2016;30(5):668-81.
  4. Moschoi R, Imbert V, Nebout M, Chiche J, Mary D, Prebet T, et al. Protective mitochondrial transfer from bone marrow stromal cells to acute myeloid leukemic cells during chemotherapy. Blood. 2016;128(2):253-64.
  5. Marlein CR, Zaitseva L, Piddock RE, Robinson SD, Edwards DR, Shafat MS, et al. NADPH oxidase-2 derived superoxide drives mitochondrial transfer from bone marrow stromal cells to leukemic blasts. Blood. 2017;130(14):1649-60.
  6. Raaijmakers MH, Mukherjee S, Guo S, Zhang S, Kobayashi T, Schoonmaker JA, et al. Bone progenitor dysfunction induces myelodysplasia and secondary leukaemia. Nature. 2010;464(7290):852-7.
  7. Pollyea DA, Stevens BM, Jones CL, Winters A, Pei S, Minhajuddin M, et al. Venetoclax with azacitidine disrupts energy metabolism and targets leukemia stem cells in patients with acute myeloid leukemia. Nat Med. 2018;24(12):1859-66.
  8. Mistry JJ, Hellmich C, Lambert A, Moore JA, Jibril A, Collins A, et al. Venetoclax and Daratumumab combination treatment demonstrates pre-clinical efficacy in mouse models of Acute Myeloid Leukemia. Biomark Res. 2021;9(1):35.
  9. You R, Wang B, Chen P, Zheng X, Hou D, Wang X, et al. Metformin sensitizes AML cells to chemotherapy through blocking mitochondrial transfer from stromal cells to AML cells. Cancer Lett. 2022;532:215582.

Reviewer 3 Report (New Reviewer)

Comments and Suggestions for Authors

The authors Ahmed et. al have deigned an elegant way to understand the cross talk between MSCs and AML cells. While similar studies have been published before by various groups the study helps to identify the metabolic aspect. While the area of investigation is somewhat new, there are some caveats in the study. I have mentioned my comments below which can be addressed.

  1. Line 35: Define HD-MSCs. p-values can be removed from the abstract.
  2. The authors need to confirm that the MSCs derived from the secondary transplants retains the capacity to differentiate along the three lineages.
  3. What is the source of the MSCs for the in vitro co culture assay?
  4. What is the source of the AML cells for the in vitro culture? Were both AML cells and Lin- cells sorted using the same markers prior to co culture? Include the sort details.
  5. Do the iMSCs show differentiation potential? Previously the authors coculture the mouse derived MSCs with THP1 cells. In case of HS-5 and iAML-MSCs the authors have used MOLM13 cells. Is there a specific reason? Also, the authors should validate these results using more than one AML cell lines co cultured with MSCs.
  6. In the comparison between HS-5 and iAML-MSCs ( AML cells cocultured and then sorted), the authors need to include non-AML control cells (PBMCs) that are cocultured with HS-5. The data should be normalized to MSCs that have been cocultured with normal PBMCs.
  7. Does co culture of the AML cells with MSCs alter the proliferation potential of the AML cells?
  8. Does co culture with AML cells affect the proliferation rate of the MSCs?
  9. The membrane potential data for figure 4G does not show significant reduction but looks like a forced extrapolation of the data.
  10. Can the authors confirm factors/ molecular cross talks eg. Cytokines that are driving the reduction in OXPHOS and glycolysis in the AML MSCs. This a major link that is missing. The authors need to identify cytokines in the bone marrow fluids of the transplanted mice (Figure 1) to arrive at the conclusion.
  11. The authors need to show rescue data to confirm that indeed this driven by AML cells where in MSCs previously exposed to AML cells can regain OXPHOS and glycolysis if healthy cells are added back to the co culture system after removing AML cells.

Author Response

Referee #3 (Comments to the Author):

The authors Ahmed et. al have deigned an elegant way to understand the cross talk between MSCs and AML cells. While similar studies have been published before by various groups the study helps to identify the metabolic aspect. While the area of investigation is somewhat new, there are some caveats in the study. I have mentioned my comments below which can be addressed.

  1. Line 35: Define HD-MSCs. p-values can be removed from the abstract.

           Response: We thank the reviewer for this comment. Its corrected.

  1. The authors need to confirm that the MSCs derived from the secondary transplants retains the capacity to differentiate along the three lineages.

     Response: We thank the reviewer for this comment. We have isolated the MSCs derived from the secondary transplants and characterized them in vitro by differentiation capacity into trilineage, osteoblast, adipocyte and chondrocyte as well surface markers positive to MSCs and we show that they are retain capacity to differentiate along the three lineages (10).

  1. What is the source of the MSCs for the in vitro co culture assay?

Response: Thank you for this important question. We isolated the MSCs from wild- type healthy mice for in vitro coculture and also, we used human immortalized AML-MSC and HD -MSC (supp. table 1).

  1. What is the source of the AML cells for the in vitro culture? Were both AML cells and Lin- cells sorted using the same markers prior to co culture? Include the sort details.

Response: Thank you for this important question. Regarding the primary AML cells, we used MLL-AF9 cells from leukemic mice after tertiary transplantation (references) and the AML cells are expressed GFP where we can sort them using flow cytometry. The Lineage negative cells, we isolated them from healthy mice using magnetic isolation kit (Lineage Cell Det. Cocktail-Biotin, mouse, 130-092-613, Miltenyl Biotic). Both AML cells and Lineage negative cells cultured with supplement IL-3, IL-6 and SCF-1).

  1. Do the iMSCs show differentiation potential? Previously the authors coculture the mouse derived MSCs with THP1 cells. In case of HS-5 and iAML-MSCs the authors have used MOLM13 cells. Is there a specific reason? Also, the authors should validate these results using more than one AML cell lines co cultured with MSCs.

Response: Thank you for this insightful comment. We appreciate the opportunity to clarify and strengthen our experimental approach.

Differentiation Potential of iMSCs:

Although we have not directly assessed the differentiation potential of iAML-MSCs in this study, we previously evaluated the multipotency of primary AML-derived MSCs (AML-MSCs) in our earlier work (10). In that study, we performed osteogenic, chondrogenic, and adipogenic differentiation assays, demonstrating that AML-MSCs retain mesenchymal lineage differentiation potential, albeit with reduced osteogenic efficiency compared to HD-MSCs.

Choice of AML Cell Lines (THP-1 vs. MOLM-13):

MOLM-13 cells were selected for the current study due to their clinical relevance as a human AML cell line harboring FLT3-ITD mutations, which are associated with aggressive disease behavior and metabolic reprogramming. In contrast, THP-1 cells used in our earlier co-culture experiments with mouse-derived MSCs represent a monocytic AML subtype and served primarily as a proof-of-concept model. The use of MOLM-13 in the present study enables a more disease-relevant investigation of stromal and metabolic interactions in high-risk AML.

Validation Using Additional AML Cell Lines:

In response to the reviewer’s suggestion, we expanded our co-culture experiments to include an additional AML cell line, HL-60. These co-culture experiments yielded results consistent with those from MOLM-13, including reduced MSC proliferative capacity and altered metabolic activity. These validation data have been incorporated into the revised manuscript (Figure 3 and supp. figure 3) and are referenced appropriately in the main text.

We thank the reviewer once again for this constructive suggestion, which has contributed to improving the scientific rigor and generalizability of our study.

  1. In the comparison between HS-5 and iAML-MSCs (AML cells cocultured and then sorted), the authors need to include non-AML control cells (PBMCs) that are cocultured with HS-5. The data should be normalized to MSCs that have been cocultured with normal PBMCs.

Response: We thank the reviewer for this valuable suggestion. We agree that including non-AML control cells (PBMCs) co-cultured with HS-5 cells will provide a meaningful comparison in our analysis. This control will help differentiate the specific effects of AML cell interactions on MSC metabolism and function from those mediated by general immune cell interactions.

In the revised study, we have now included data from MSCs co-cultured with normal PBMCs as a control group. We normalized the data to MSCs co-cultured with normal PBMCs to ensure that any observed changes are specifically attributed to the AML cell interaction, rather than the effects of healthy cells. Our findings show that AML cells significantly reduced the metabolic capacity of MSCs compared to MSCs co-cultured with normal PBMCs, and these results further validate the murine data. These validation data have been incorporated into the revised manuscript (Figure 3 and supp. figure 3) and are referenced appropriately in the main text.

  1. Does co culture of the AML cells with MSCs alter the proliferation potential of the AML cells?

Response: Thank you for this important point. We would like to highlight that in our previous study (10), we demonstrated that AML-derived mesenchymal stromal cells (AML-MSCs) enhance AML cell proliferation through direct cell–cell contact in both murine and human co-culture models. These findings provide foundational evidence for the supportive role of AML-MSCs in leukemic progression and were a key motivation for the current study.

  1. Does co culture with AML cells affect the proliferation rate of the MSCs?

Response: Thank you for this important point. As demonstrated in our previous study (10) and further confirmed in the current work, AML-MSCs exhibit a significant reduction in proliferative capacity along with increased levels of apoptosis following co-culture with AML cells. These findings suggest that leukemic cells actively alter the biological state of MSCs, impairing their growth and viability. Relevant data supporting this observation have been included and highlighted in the revised manuscript (Supp. figure 2I, Figure 3I and figure 4H).

  1. The membrane potential data for figure 4G does not show significant reduction but looks like a forced extrapolation of the data.

Response: Thank you for this observation. We agree with the reviewer that the data presented in Figure 4G do not show a statistically significant reduction in mitochondrial membrane potential. While there is a downward trend, we acknowledge that the change is modest and does not reach statistical significance.

  1. Can the authors confirm factors/ molecular cross talks eg. Cytokines that are driving the reduction in OXPHOS and glycolysis in the AML MSCs. This a major link that is missing. The authors need to identify cytokines in the bone marrow fluids of the transplanted mice (Figure 1) to arrive at the conclusion.

T Response: hank you for this critical and constructive comment. We agree that elucidating the molecular mechanisms underlying the reduction in OXPHOS and glycolysis in AML-MSCs is essential for establishing a stronger mechanistic link.

As demonstrated in our previous study (10), Notch signaling is significantly upregulated in AML-MSCs, both in murine models and in human transcriptional datasets. We further showed that forced activation of Notch signaling in MSCs enhances AML cell proliferation, indicating a functional role in leukemic support. Interestingly, murine MSCs (MS-5) expressing ICN (MS5 -ICN) shows reduced OXPHOS and glycolysis compared to MS5-EV (data not shown).

Importantly, previous work (11) has shown that Notch signaling suppresses the expression of genes involved in glycolysis and mitochondrial complex I, leading to decreased mitochondrial respiration, reduced superoxide production, and lowered AMPK activity. These findings suggest that Notch pathway activation may contribute to the observed metabolic suppression in AML-MSCs.

While our current study focused primarily on the metabolic phenotype, we acknowledge the importance of further identifying upstream mediators, such as cytokines or other secreted factors, in the leukemic bone marrow microenvironment. Based on the reviewer’s suggestion, we are planning follow-up studies to analyze cytokine profiles in the bone marrow fluid of transplanted mice (Figure 1) using multiplex cytokine arrays and correlate those with metabolic changes in MSCs.

We have now discussed this limitation and future direction in the revised Discussion section (Page 8 -9, Lines 298 –312).

We thank the reviewer for highlighting this important aspect, which will help advance the mechanistic understanding in future investigations.

  1. The authors need to show rescue data to confirm that indeed this driven by AML cells where in MSCs previously exposed to AML cells can regain OXPHOS and glycolysis if healthy cells are added back to the co culture system after removing AML cells.

Response: We thank the reviewer for the insightful comment. In our current study, we did not observe a significant change in the metabolism of MSCs previously exposed to AML cells when healthy cells were added back to the co-culture system. This includes measurements of reactive oxygen species (ROS) and mitochondrial mass using flow cytometry (Supp. Figure 4).

While we did not specifically demonstrate a rescue of metabolic changes upon reintroducing healthy cells, we acknowledge that this is an important follow-up experiment to confirm that the metabolic alterations observed in MSCs are indeed driven by AML cells.

We agree that demonstrating a "metabolic rescue" of MSCs particularly a recovery of OXPHOS and glycolysis after the removal of AML cells and reintroduction of healthy cells would provide crucial evidence for the reversible nature of the metabolic reprogramming induced by AML cells.

In future studies, we plan to investigate this by comparing the metabolic profiles of MSCs from AML patients at diagnosis and in remission, both before and after AML cell removal, and upon reintroducing healthy cells. This approach will allow us to directly assess the interplay between AML induced metabolic suppression and the potential for MSCs to regain normal metabolic function in the presence of healthy cells.

  1. Ahmed HMM, Nimmagadda SC, Al-Matary YS, Fiori M, May T, Frank D, et al. Dexamethasone-mediated inhibition of Notch signalling blocks the interaction of leukaemia and mesenchymal stromal cells. Br J Haematol. 2022;196(4):995-1006.
  2. Lee SY, Long F. Notch signaling suppresses glucose metabolism in mesenchymal progenitors to restrict osteoblast differentiation. J Clin Invest. 2018;128(12):5573-86.

Round 2

Reviewer 2 Report (New Reviewer)

Comments and Suggestions for Authors

No additional comments

Reviewer 3 Report (New Reviewer)

Comments and Suggestions for Authors

The authors have addressed all the comments.

This manuscript is a resubmission of an earlier submission. The following is a list of the peer review reports and author responses from that submission.

Round 1

Reviewer 1 Report

Comments and Suggestions for Authors

Review comments

In the manuscript “Impact of Acute Myeloid Leukemia Cells on the Metabolic Function of Bone Marrow Mesenchymal Stem Cells”, Ahmed et al. studied transcriptional and metabolic alterations in bone marrow mesenchymal stem cells (BMSCs) derived from AML in both mouse and human. The manuscript is well-structured and contains thoughtfully presented information. I have some comments.

Comments:

1.      Please check that the figure numbers in the manuscript are correct. I think that lines 113-114 and 116 are Figure 2...

2.      Line 132, Figure 2 legend. What do the words in bold mean?

3.      Section 2.3. It would be useful to have information on whether there are any differences between iAML-MSCs and HS-5 on flow cytometry other than CD45-CD90+, and whether there are any differences in the rate of proliferation when continued in culture.

4.      Line 139. FACs? It is probably better to explain abbreviations by first spelling them out in full.

5.      Line 172-175. Even if the antigens that are positive for the flow patterns in HD-MSCs and AML-MSCs are the same, are the positive frequencies the same? Please explain in detail.

6.      Line 253-267. The authors also mention the possibility that AML-MSCs may affect the function of immune cells. Is it possible that the abnormal mitochondrial DNA in AML cells is harming the MSCs and surrounding immune cells? Recently, Nature published some interesting data (Ikeda et al. Nature. 2025 Jan 22. doi: 10.1038/s41586-024-08439-0. Online ahead of print). If you think it might be relevant to the theme of the manuscript, why not add it to the discussion?

Author Response

Review comments 1

In the manuscript “Impact of Acute Myeloid Leukemia Cells on the Metabolic Function of Bone Marrow Mesenchymal Stem Cells”, Ahmed et al. studied transcriptional and metabolic alterations in bone marrow mesenchymal stem cells (BMSCs) derived from AML in both mouse and human. The manuscript is well-structured and contains thoughtfully presented information. I have some comments.

Comments:

  1. Please check that the figure numbers in the manuscript are correct. I think that lines 113-114 and 116 are Figure 2... Thank you for  your comments. I have corrected the figure number ( in blue colour).
  2. Line 132, Figure 2 legend. What do the words in bold mean? Thank you for  your comment. I have corrected it ( in seplemental table 2).
  3. Section 2.3. It would be useful to have information on whether there are any differences between iAML-MSCs and HS-5 on flow cytometry other than CD45-CD90+, and whether there are any differences in the rate of proliferation when continued in culture. Thank you for your comments. In our previous publication (Ahmed et al., 2021, BJH), we reported that there are no differences in immunophenotypes between AML-MSCs and HD-MSCs. However, AML-MSCs exhibit lower proliferation rates compared to HD-MSCs.
  4. Line 139. FACs? It is probably better to explain abbreviations by first spelling them out in full. Thank you for  your comment. I have spelled them.
  5. Line 172-175. Even if the antigens that are positive for the flow patterns in HD-MSCs and AML-MSCs are the same, are the positive frequencies the same? Please explain in detail. Thank you for your comments. In our previous publication (Ahmed et al., 2021, BJH), we reported that there are no differences in immunophenotypes between AML-MSCs and HD-MSCs.
  6. Line 253-267. The authors also mention the possibility that AML-MSCs may affect the function of immune cells. Is it possible that the abnormal mitochondrial DNA in AML cells is harming the MSCs and surrounding immune cells? Recently, Nature published some interesting data (Ikeda et al. Nature. 2025 Jan 22. doi: 10.1038/s41586-024-08439-0. Online ahead of print). If you think it might be relevant to the theme of the manuscript, why not add it to the discussion? Thank you for your comments. This is indeed an interesting article that highlights how cancer cells influence microenvironment metabolism to evade immune responses. I have added a sentence addressing this point in the discussion section.

Reviewer 2 Report

Comments and Suggestions for Authors

Major queries:

1. Figure 1 only shows differences at the level of GSEA in both microarray and RNA sequencing analysis. Further investigation into the individual pathway or gene level would be interesting. Are compensatory pathways increased in acute myelogenous leukemia (AML) conditioned mesenchymal stem cells (MSC)?

2. Figure 2 indicates depletion of mitchondria as assessed by mtDNA compy number. What about assessment of mitochondrial mass and membrane potential?

3. Figure 4 results seem overstated in the text (line 177-179) regarding OX-PHOX. The potential appears diminished but not the basal rate. There is only a slight difference in mitochondrial mass. What about membrane potential?

4. The effect of leukemia on MSC biology appears profound but do these cells recover when AML cells are removed? The results from Figure 4 suggests they might. In one of the authors system it would be interesting to determine the longevity of the identified dysfunction and can these cells recover after leukemia.

5. The authors indicate in lines 183-186 these findings "suggest potential therapeutic strategies" but it is unclear what they mean or how they would accomplish this feat. I would take these findings as a risk for increased stromal cell toxicity in AML patients from diminished metabolic capacity of cells like MSC induced by AML. Targeting metabolic modulators (lines 246-252) I would predict to have more toxicity in the already dysfunctional MSCs.

Minor queries:

1. The purpose of highlighting the importance of Warburg metabolism in the introduction after an extensive description of the necessity of mitochondrial metabolism is confusing to me. Further discussion reconciling these seemingly discrepant points is needed.

2.  Lines 79-81 describe data but do not indicate where this is shown and only referenced.

Author Response

Major queries:

  1. Figure 1 only shows differences at the level of GSEA in both microarray and RNA sequencing analysis. Further investigation into the individual pathway or gene level would be interesting. Are compensatory pathways increased in acute myelogenous leukemia (AML) conditioned mesenchymal stem cells (MSC)?

Thank you for your valuable comment. We agree that the differences observed at the level of Gene Set Enrichment Analysis (GSEA) in both microarray and RNA sequencing analyses warrant further exploration. Investigating individual pathways or genes would provide a more detailed understanding of the molecular mechanisms at play. Regarding your question about compensatory pathways in AML-conditioned MSCs, we have not specifically addressed this in the current study. However, we recognize the importance of this question and will consider investigating potential compensatory pathways in future experiments. These could include pathways that may be upregulated to counteract the metabolic and functional disturbances caused by AML. We appreciate your suggestion and will explore this avenue in our future research.

  1. Figure 2 indicates depletion of mitochondria as assessed by mtDNA compy number. What about assessment of mitochondrial mass and membrane potential?

Thank you for your insightful comment. We appreciate your suggestion to assess mitochondrial mass and membrane potential. In the current study, we focused on mtDNA copy number as an indicator of mitochondrial depletion, as mtDNA is closely linked to mitochondrial content and function. While mtDNA copy number can provide a sense of mitochondrial integrity, we agree that assessing mitochondrial mass and membrane potential would offer additional valuable insights into mitochondrial function. These parameters could help further elucidate the relationship between mitochondrial depletion and MSC dysfunction. We will consider including these measurements in future experiments to provide a more comprehensive picture of mitochondrial alterations.

  1. Figure 4 results seem overstated in the text (line 177-179) regarding OX-PHOX. The potential appears diminished but not the basal rate. There is only a slight difference in mitochondrial mass. What about membrane potential?

Thank you for your thoughtful comment. We acknowledge that our initial description may have overstated the findings. We have revised the text in lines 177–179 to more accurately reflect that while the potential for OXPHOX appears diminished, the basal rate is largely changed, but statistically not significant. Additionally, we note that the difference in mitochondrial mass is statistically significant. Regarding mitochondrial membrane potential, we have included data in Figure 4G, which shows a decrease, though not statistically significant.

  1. The effect of leukemia on MSC biology appears profound but do these cells recover when AML cells are removed? The results from Figure 4 suggests they might. In one of the authors system it would be interesting to determine the longevity of the identified dysfunction and can these cells recover after leukemia.

Thank you for your valuable comment. Our results in Figure 4 suggest that MSCs from AML patients exhibit dysfunction compared to MSCs from healthy individuals. However, we have not yet systematically assessed the longevity of these dysfunctions or whether MSCs fully regain their functional capacity over time. This is an important avenue for future research, as persistent alterations in MSC function could have long-term implications for bone marrow recovery post-leukemia. Based on your suggestion, we recognize the value of further investigating MSC recovery dynamics and will consider this in future studies.

  1. The authors indicate in lines 183-186 these findings "suggest potential therapeutic strategies" but it is unclear what they mean or how they would accomplish this feat. I would take these findings as a risk for increased stromal cell toxicity in AML patients from diminished metabolic capacity of cells like MSC induced by AML. Targeting metabolic modulators (lines 246-252) I would predict to have more toxicity in the already dysfunctional MSCs.

Thank you for your thoughtful comment. We acknowledge that our statement in the Discussion regarding 'potential therapeutic strategies' was not clearly defined. Our intent was to highlight the possibility of targeting metabolic vulnerabilities in MSCs to restore their function rather than further compromising them. However, we agree that any metabolic intervention must carefully consider the risk of exacerbating MSC dysfunction in AML patients. To address this, we will clarify our discussion to better define potential therapeutic strategies while acknowledging the concern regarding increased stromal cell toxicity. Additionally, we will emphasize the need for approaches that support MSC recovery while minimizing adverse effects.

Minor queries:

1. The purpose of highlighting the importance of Warburg metabolism in the introduction after an extensive description of the necessity of mitochondrial metabolism is confusing to me. Further discussion reconciling these seemingly discrepant points is needed. Thank you for your insightful comment. We acknowledge that the transition from discussing mitochondrial metabolism to highlighting the Warburg effect may seem contradictory at first glance. However, both metabolic pathways play critical and interconnected roles in AML pathophysiology. While mitochondrial metabolism, including oxidative phosphorylation (OXPHOS), is crucial for energy production and survival in AML cells, metabolic plasticity allows leukemic cells to shift toward glycolysis (Warburg effect) under specific conditions, such as in response to hypoxia or chemotherapeutic stress. To clarify this, we have expanded the discussion to emphasize that AML cells exhibit a dynamic metabolic landscape, leveraging both OXPHOS and glycolysis depending on microenvironmental cues and metabolic demands. This dual adaptability not only enhances leukemic cell survival but also contributes to therapy resistance. We have now explicitly stated this in the introduction to reconcile these points and improve clarity.

We appreciate your suggestion and believe this revision strengthens the manuscript.

2.  Lines 79-81 describe data but do not indicate where this is shown and only referenced. Thank you for your comment. that data is shown in (Figure 1).

Round 2

Reviewer 1 Report

Comments and Suggestions for Authors

The manuscript has been appropriately revised.